# Environmental characteristics around the household and their association with hookworm infection in rural communities from Bahir Dar, Amhara Region, Ethiopia

Melaku Anegagrie[1,2☉], Sofía Lanfri[3,4☉], Aranzazu Amor Aramendia[1,2], Carlos Matías Scavuzzo[3,5], Zaida Herrador[2], Agustín Benito[2], Maria Victoria Periago[4]*

**1** Fundación Mundo Sano, Madrid, Spain, **2** National Centre for Tropical Medicine, Institute of Health Carlos III, Madrid, Spain, **3** Instituto de Altos Estudios Espaciales Mario Gulich, Comisión Nacional de Actividades Espaciales, Universidad Nacional de Córdoba, Córdoba, Argentina, **4** Fundación Mundo Sano, Buenos Aires, Argentina, **5** Consejo Nacional de Investigaciones Científicas y Técnicas (CONICET), Buenos Aires, Argentina

☉ These authors contributed equally to this work.
* vperiago@mundosano.org

## Abstract

Soil-Transmitted Helminths (STH) are highly prevalent Neglected Tropical Disease in Ethiopia, an estimated 26 million are infected. Geographic Information Systems and Remote Sensing (RS) technologies assist data mapping and analysis, and the prediction of the spatial distribution of infection in relation to environmental variables. The influence of socioeconomic, environmental and soil characteristics on hookworm infection at the individual and household level is explored in order to identify spatial patterns of infection in rural villages from Zenzelema (Amhara region). Inhabitants greater than 5 years old were recruited in order to assess the presence of STH. Socioeconomic and hookworm infection variables at the household level and environmental variables and soil characteristics using RS were obtained. The dominant STH found was hookworm. Individuals which practiced open defecation and those without electricity had a significant higher number of hookworm eggs in their stool. Additionally, adults showed statistically higher hookworm egg counts than children. Nonetheless, the probability of hookworm infection was not determined by socioeconomic conditions but by environmental characteristics surrounding the households, including a combination of vigorous vegetation and bare soil, high temperatures, and compacted soils (high bulk density) with more acidic pH, given a pH of 6.0 is optimal for hatching of hookworm eggs. The identification of high-risk environmental areas provides a useful tool for planning, targeting and monitoring of control measures, including not only children but also adults when hookworm is concerned.

**Data Availability Statement:** All relevant data are within the manuscript and its Supporting Information files.

**Funding:** This study was funded by Fundación Mundo Sano and Instituto de Salud Carlos III. CMS has a PhD scholarship from Consejo Nacional de Investigaciones Científicas y Técnicas (CONICET). The funders had no roles in the design of the study or collection, analysis and interpretation of the data.

**Competing interests:** The authors have declared that no competing interests exist.

## Author summary

Soil-Transmitted Helminths (STH) are a group of intestinal parasites that are included in the list of Neglected Tropical Diseases (NTDs) elaborated by the World Health Organization (WHO). This group includes roundworms, whipworms, and hookworms. Sub-Saharan Africa (SSA) is one of the most largely affected by NTDs and Ethiopia harbours one of the largest burdens of STH, especially hookworm, with 10 million infected. In this study we aimed to explore the association between the environment, soil and socioeconomic characteristics most associated with the presence of hookworm infection in a rural area from Bahir Dar, Amhara Region, Ethiopia. Results of this study showed that the presence of hookworm around the household is associated to environmental characteristics such as high temperatures, a combination of vigorous vegetation and bare compacted soil and acidic pH. On the other hand, the intensity of hookworm infection was associated with socioeconomic conditions such as the lack of latrines with the practice of open defecation and a lack of electricity. Therefore, in order for the infection to establish itself in a community, certain environmental characteristics are needed, but once the infection is established, certain socioeconomic characteristics play a role in its transmission pattern.

## Introduction

Neglected Tropical Diseases (NTDs) are widespread in Sub-Saharan African (SSA) countries [1], and Africa carries the largest burden of NTDs worldwide, with 39% of the total burden [2]. Ethiopia is not an exception and it is one of the countries with the highest burden of disease in SSA, with an estimated 80 million people living in NTD endemic areas out of a population of 115 million [3]; with the presence of almost all 20 diseases on the list, with the exception of Chagas Disease and Yaws [4]. The Soil-Transmitted Helminths (STH) are the most prevalent NTD in the country, with an estimated 81 million people living in STH endemic areas [5]. Moreover, the presence of *Strongyloides stercoralis* has also been detected [6–9], but the actual burden is unknown since the techniques usually used for diagnosis of STH are not sensitive enough for this parasite [7].

In 2004, Ethiopia launched a deworming strategy focused on preschool aged children (PSAC) that was implemented in every district of the country, except the capital city of Addis Ababa [4]. The program was organized by the Regional Health Bureaus (RHB) and by 2009; it had reached 11 million children [4]. In 2013, the Federal Democratic Republic of Ethiopia Ministry of Health (FMHO) launched its first NTD Master Plan and established a NTD Case Team within the Ministry [10]. National guidelines for STH and *Schistosoma* were put in place in 2014 [11] and in 2015 a national deworming program for school aged children (SAC) was launched [11], with a revision and update of the NTD Master Plan in 2016 [5]. The number of people living in endemic areas for STH was approximately 79 million and the population living in areas that would qualify for mass drug administration (MDA) is comprised of 4.6 million PSAC, 17.7 million SAC and 31.3 million adults [5,12]. A total of 476 woredas (districts) were identified through these different studies as in need of MDA for STH with prevalences ranging from 20% to greater than 50%, so that 279 woredas would require treatment twice per year according to World Health Organization (WHO) guidelines [11]. The regional states most affected were Amhara, Gambela, Southern Nations, Nationalities and Peoples (SNNP) and the western part of Oromia.

Understanding the distribution of STH in specific areas as well as the factors most associated with their presence would allow tailoring programs to not only identify those areas most

likely to be affected for allocation of resources but also to promote health practices to avoid reinfection after treatment. Geographic Information Systems (GIS) and Remote Sensing (RS) technologies assist not only data mapping and analysis but also the prediction of the spatial distribution of infection in relation to RS and environmental variables [13]. Maps specific for STH have been developed in different countries in order to determine areas at risk for the presence of these parasites, most of them using historical prevalence data from either point studies [14–17] or national STH data [18], but none of these were conducted at the household level. Most studies conducted at the household level have determined certain risk factors for infection with STH, but environmental parameters were not included [19–22].

Previous studies conducted in Amhara have shown that hookworm is the predominant species of STH present in rural areas [6–9,23–25]. Given that Ethiopia is the country with third highest burden of hookworm in SSA, in the current study we explore the influence of socioeconomic, environmental and soil characteristics on the infection of hookworm at the individual and household level in order to identify spatial patterns of infection in a community from a rural *kebele* of Amhara region, Ethiopia. Highlighting the identification of high-risk areas could provide a very useful tool for planning, targeting and monitoring of control measures.

## Methods

### Ethics statement

The Amhara National Regional State Health Bureau Ethics Review Committee revised and approved the study protocol (Ref. n˚: 1/87/2008). According to the principles of the Helsinki Declaration, informed consent for stool examination was sought, as well as withdrawal, guarantee of anonymity, treatment and follow up. Written informed consent was set for adults and children; for the second group, parents or guardians signed the consent form.

**Study area.**  Ethiopia is a Federal Democratic Republic comprised of ten regional states and two administrative cities which are then subdivided into 95 zones and 839 administrative *woredas* or districts. *Woredas* are further divided into 16,253 *kebeles*, the smallest administrative unit. Most of the country is suitable for the transmission of STH. The current study was conducted in the Amhara Region which is located in the north-western part of the country. The study was carried out in the rural *kebele* of Zenzelema, about 20 km east of Bahir Dar City, the capital of Amhara, in the South shore of Tana Lake (Fig 1). The district is located in the highlands, at 1,700 to 2,000 m above sea level. The area is under the influence of three seasons: rainy, from June to September; spring, from October until January; and a dry season the rest of the year. This *kebele* is made of nine *gotts* or villages; five of them were randomly selected for the study [8].

Zenzelema, located in the main road, with poor access to water, Mazoria and Sesaberet which are accessible by walking through earth tracks and have smaller streams and watercourses with natural water, and Kurbi and Gedro, located in more remote areas near Tana Lake, with no access tracks. In all these villages, except for Zenzelema, the houses are far apart from one another and surrounded by crops and wooded areas. On the other hand, Zenzelema is a congested settlement where houses are adjacent to one another (Fig 2). The family subsistence economy is based on small cultivations and livestock (mainly cattle, sheep and goats). Even though the area is on the shore of Blue Nile and Tana Lake, water resources for cultivation are markedly dependent on the rain, with no existing infrastructure for rainwater collection. Moreover, land degradation because of erosion, mostly related to high deforestation rates, is common, with high amounts of soil lost every year.

*Data collection.* The data of this study were collected in the frame of a larger research aimed at determining the prevalence of the STH *S. stercoralis* at the school and community level in an

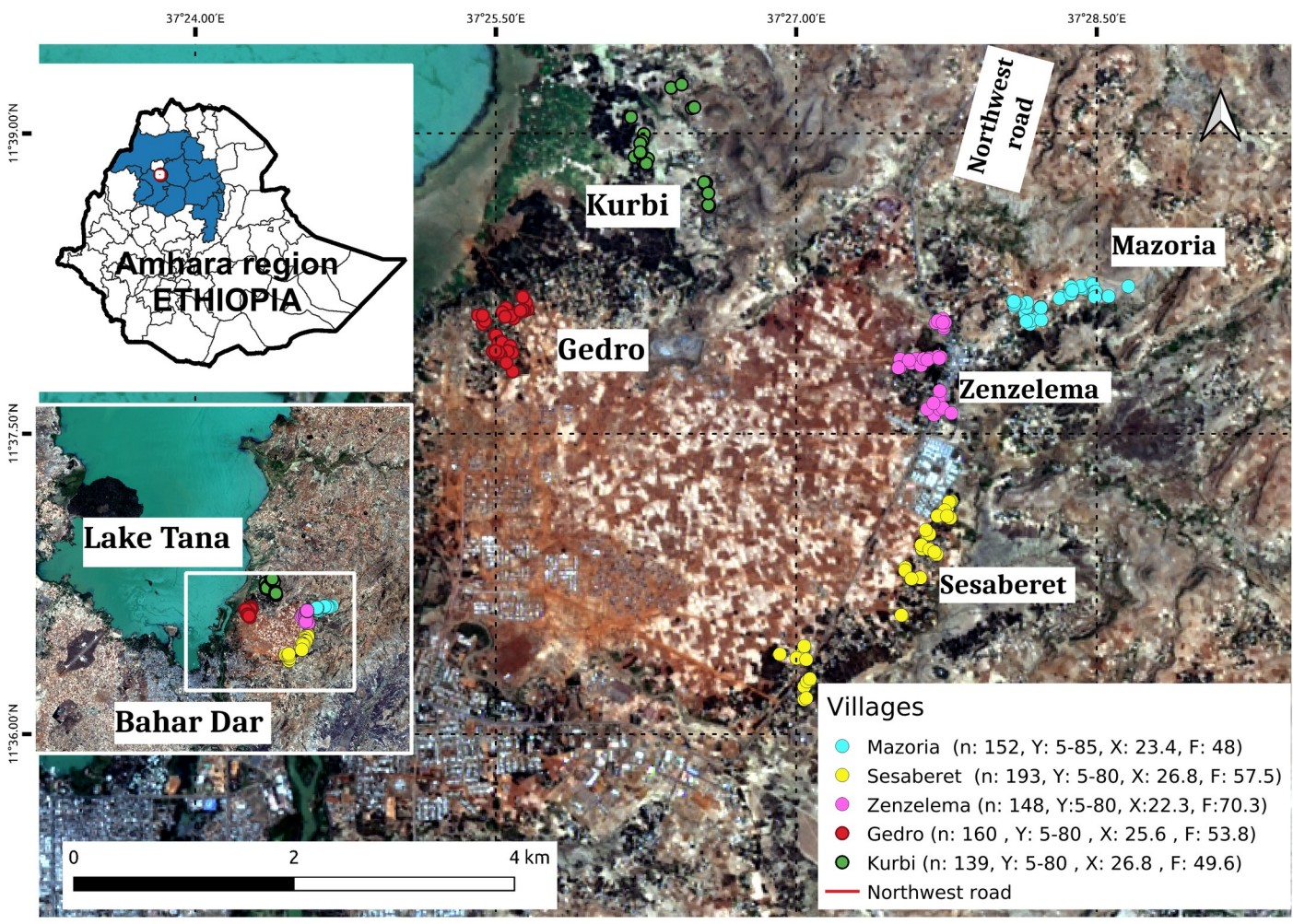

**Fig 1. Study area in Amhara Region, Ethiopia.** The villages included in this study were Kurbi, Gedro, Mazoria, Sesaberet and Zenzelema, belonging to a rural district. The number of people (n) recruited per village as well as their age range in years (Y), average age (X) and distribution by sex (F = % female) are noted in the box on the right hand corner. Landsat-8 images, "Image courtesy of the U.S. Geological Survey", downloaded from the US Geological Survey server (USGS, https://www.usgs.gov) using Earth Explorer platform (https://earthexplorer.usgs.gov). The geographical location for Landsat-8 is: Path 170, Row 52. DATE_ACQUIRED = 2016-03-10.

area known to be of high prevalence for hookworm [7,8]. Briefly, all inhabitants over 5 years of age, living in the area for at least three months, were invited to participate. Information about individual (children and adult) and household characteristics was obtained by using standardized surveys, based on the WHO/UNICEF Joint Monitoring Program for Water Supply, Sanitation and Hygiene [26] adapted to the Ethiopian culture and translated into Amharic language, including sociodemographic and socioeconomic data. All participants were asked to answer an individual questionnaire while the head of the family was also asked to answer a household questionnaire. From February to June 2016, a field team composed of a nurse, two HEW and the project coordinator went house to house, to collect the samples, conduct the questionnaires and georeference the households. The hookworm infection prevalence and intensity was obtained from the study previously published [8] in which stool samples were processed by three coprological techniques: formalin-ether concentration (FEC), Baermann technique (BT) and McMaster counting technique (MM).

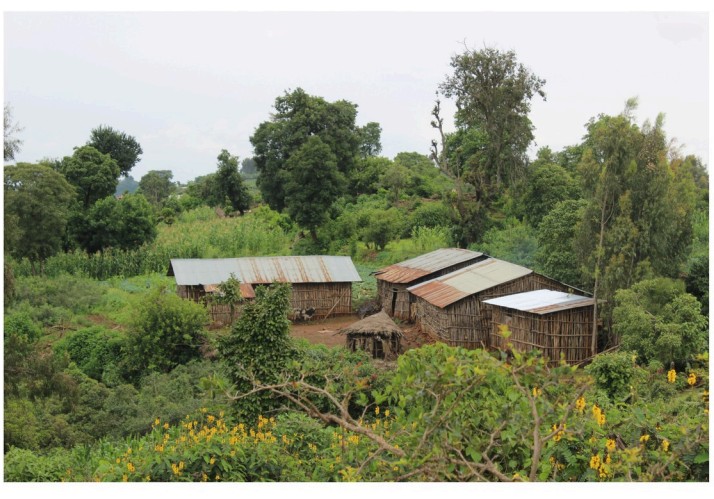
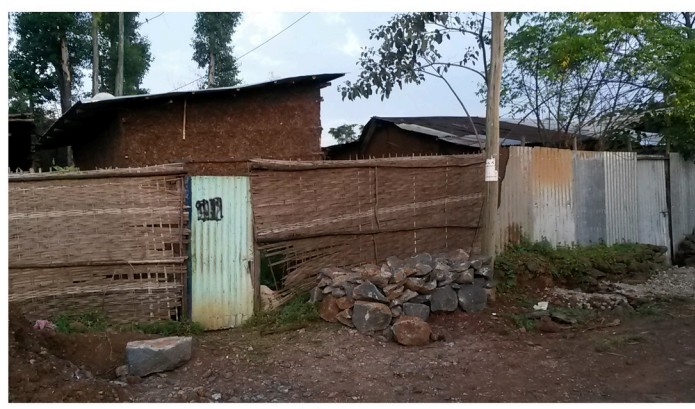

A B

**Fig 2.** An example of the villages included in the study with Mazoria (A) as an example of a more rural and isolated village and Zenzelema (B) as an example of a more urbanized village in the main road.

**Household characterization.** Through the different questionnaires and sample collection, socioeconomic and hookworm infection variables were obtained and analysed at the household level using a geographic information system (QGIS software). Several socioeconomic variables were measured related to the number of individuals per household, income, toilet type, water source, presence of electricity, etc. These are further detailed in the result section. The parasitological parameters used for hookworm were the number of hookworm infected individuals per household, the individual EPGs and the geometric mean number of EPG per household (geometric mean EPG of infected individuals per household).

**Soil and Environmental characterization.** The environmental variables selected for analysis were those related to the variables reported as significant in previous studies that focused on the risk of STH infections [14–18,21,27–33]. The selected environmental variables obtained through remote sensors were precipitation, temperature, NDVI, NDBI, MSI and BSI. These are listed in Table 1, together with their source, characteristics and indexes used to calculate them.

**Table 1. Environmental variables used with their characteristics, sources and indexes used to calculate them.**

| Environmental Variables | Source | Characteristics/ Sentinel-2A Indexes |
|---|---|---|
| Precipitation | WorldClim version 2.1 climate data for 1979–2000 (New version released in January 2020) | Spatial resolution of 30 seconds (~1 km2)/ NA |
| Temperature | WorldClim version 2.1 climate data for 1979–2000 (New version released in January 2020) | Mean temperature from monthly January/ NA |
| Normalized Difference Vegetation Index (NDVI) | Satellite imagery Sentinel-2A of 01 Feb 2017* Level 1C product | Numerical indicator highly associated with vegetation content/ (B8 –B4) / (B8 + B4) |
| Normalized Difference Built-up Index (NDBI) | Satellite imagery Sentinel-2A of 01 Feb 2017* Level 1C product | Highlights urban areas/ (B11-B8)/(B11+B8) |
| Moisture Stress Index (MSI) | Satellite imagery Sentinel-2A of 01 Feb 2017* Level 1C product | Index of hydric stress/ (B11 / B08) |
| Bare Soil Index (BSI) | Satellite imagery Sentinel-2A of 01 Feb 2017* Level 1C product | Numeric indicador that enhances the identification of bare soil areas and fallow lands/ (B11 + B4)–(B8 + B2) / (B11 + B4) + (B8 + B2) |

*Note: Source of ESA, image courtesy of the U.S. Geological Survey. Free Sentinel-2A imagery was downloaded from the ESA Sentinel data hub (https://scihub.copernicus.eu).

With respect to the interpretation of these indexes, high NDVI values correspond to areas that reflect more in the near-infrared spectrum, which indicates denser and healthier vegetation [34]. This index is related to MSI since it is a measure of hydric stress, with the highest numbers indicating greater hydric stress of plants and therefore a lower content of soil humidity [35]. The values for MSI range from 0 to greater than 3, so the typical range of this index for vegetation is around 0.2 to 2 [36]. On the other hand, NDBI highlights urban areas and is very helpful for estimating surfaces with buildings with respect to bare areas or those with vegetation [37]. Finally, BSI is related to NDBI in the sense that it is a numeric indicator that enhances the identification of bare soil areas and fallow lands [38].

Soil variables with greater spatial resolution (250 m) were available for Ethiopia. Those variables related to the ones previously used [14]; organic carbon content, acidity, bulk density and gypsum content were chosen and are listed in Table 2. BLDFIE is the ratio of the oven-dry mass of the solids to the volume of the soil; the bulk volume includes not only the volume of the solids but also that of the pore space [39]. With respect to the pH variables of the soil, they describe more than relative acidity or alkalinity, it also provides information on nutrient availability, metal dissolution chemistry, and the activity of microorganisms [40], and therefore it might influence the presence or absence of hookworm larvae in the soil. The remaining soil variables refer to the depth of the bedrock (BDRICM) in cm to a lithic or paralithic contact and its composition, with the weight percentage of the clay particles (CLYPPT) as $< 2$ μm of the mass fraction of soil material at $< 2$ mm, and the volumetric percentage of coarse fragments (CRFVOL) as the mass fraction of the soil material at $> 2$ mm [39].

In order to determine any association with hookworm infection between any of these environmental and soil variables, given the route of infection of the parasite with a necessary passage through the soil, the average of each of these variables were estimated with a 30 m buffer radius around each household using QGIS software.

**Statistical analysis.**   SataScan was used to analyse the geographic spatial pattern of the parasitological indexes in the study area [41,42]. The presence of statistically significant spatial conglomerates was analysed, as well as that of the non-random distribution in space. This was performed by gradually scanning a window across space, noting the number of observed and expected observations inside the window at each location, and a p-value was assigned to the cluster. Scan statistics use a different probability model depending on the nature of the data. In this study, a Purely Spatial analysis was performed, with the objective of scanning for clusters with high values using the normal model for continuous data [41]. The unit of analysis was the

**Table 2. Soil variables used with their abbreviation, characteristics and unit of measurement.**

| Soil variables | Abbreviation | Characteristics (Unit) |
|---|---|---|
| Soil organic carbon content | ORCDRC | NA (permille) |
| Soil organic carbon stock | Ocstha | NA (ton/ha) |
| Bulk density | BLDFIE | fine earth (kg/m3) |
| pH index measured in KCl solution | Phikcl | Relative acidity or alkalinity |
| pH index measured in water solution | Phihox | Relative acidity or alkalinity |
| Weight percentage of the silt particles | SLTppt | Particles about 0.0002–0.05 mm (percentage) |
| Cation Exchange Capacity of soil | CECSOL | NA (cmolc/kg) |
| Depth to bedrock | BDRICM | R horizon up to 200 cm (cm) |
| Weight percentage of the clay particles. | CLYPPT | Particles about <0.0002 mm (percentage) |
| Volumetric percentage of coarse fragments. | CRFVOL | Coarse fragments about >2 mm (percentage) |

*Note: NA: Not Applicable. Source: ISRIC—World Soil Information (39).

household and the variables evaluated were geometric mean EPG per household and number of individuals infected.

The association between individual EPG and characteristics of the household was analysed using the Mann-Whitney U (MWU) and Kruskal-Wallis H (KWH) tests. Chi square test was used for comparing the prevalence of the infection between villages. The individual social and economic characteristics evaluated are listed in the results section. Estimation of the household risk of hookworm infection was performed through a linear multiple regression analysis. The number of hookworm infected individuals per household as the dependent variable was modelled in relation to the predictive variables: environmental (Table 1), soil characteristics (Table 2) and socioeconomic variables (S1 Table). Statsmodel Python module (statsmodel.org) was used for the comparison between groups as well as to adjust the regression model.

## Results

### Hookworm infection characteristics and spatial distribution

A total of 792 individuals from 241 households participated in the study by providing stool samples: 152 individuals from Mazoria, 193 from Sesaberet, 160 from Gedro, 139 from Kurbi and 148 from Zenzelema. The descriptive characteristics of hookworm infection in the five villages are detailed in Table 3. The mean number of people infected with hookworm was lower in Zenzelema (64.2%) and highest in Kurbi (87.8%), although the difference in prevalence and the mean eggs per gram of feces (EPG) per household between the villages was not statistically significant. In the larger previous study conducted in this study population [8], the presence of hookworm, *S. stercoralis*, *A. lumbricoides* and *T. trichiura* was detected, with a dominance of hookworm in Mazoria and Sesaberet, and a dominance of *S. stercoralis* in Zenzelema. With respect to hookworm, Kurbi not only had the highest prevalence of infection (87.8%) but also the greatest average of infected individuals per household (3.0) and the highest geometric mean EPG per household (1897.7 EPG). With respect to the intensity of hookworm infection, most infections were light and the only heavy intensity infection was found in Gedro (Table 3).

As far as the hookworm infection per household, the spatial distribution among the five villages is shown in Fig 3. In this figure, the data listed in Table 3 is verified. Many of the

**Table 3. Hookworm infection status of the participants of the study from the five villages, Kurbi, Gedro, Mazoria, Sesaberet and Zenzelema (Amhara State, Ethiopia).**

| Hookworm infection characteristics | Mazoria n = 152 | Sesaberet n = 193 | Gedro n = 160 | Kurbi n = 139 | Zenzelema n = 148 |
|---|---|---|---|---|---|
| Prevalence [%] | 79.6 | 78.8 | 83.8 | 87.8 | 64.2 |
| Number of infected individuals [No.] | 121 | 152 | 134 | 122 | 95 |
| No. of infected individuals by intensity as per MM [No.] | 121: 120 light 1 moderate | 150*: 144 light 6 moderate | 134: 129 light 4 moderate 1 heavy | 122: 113 light 9 moderate | 94*: 92 light 2 moderate |
| Mean number of infected individuals per household (mean) | 2.5 | 2.7 | 2.9 | 3.0 | 2.3 |
| Mean EPG per household (geometric mean and 95%CI) | 823.6 (690.3–982,7) | 916.9 (766.2–1097.4) | 915.9 (794.8–1055.3) | 1897.7 (1629.2–2210.4) | 358.7 (288.8–445.6) |

*Note: MM: MacMaster Technique. EPG: Eggs per gram of feces. CI: Confidence Interval. Intensity of infection for hookworm was measured according to WHO guidelines where: 1 to 1999 EPG is a light infection, 2000 to 3999 EPG is medium and >4000 EPG is heavy [43]. *Lack of MM data for two samples from Sesaberet and one sample from Zenzelema.

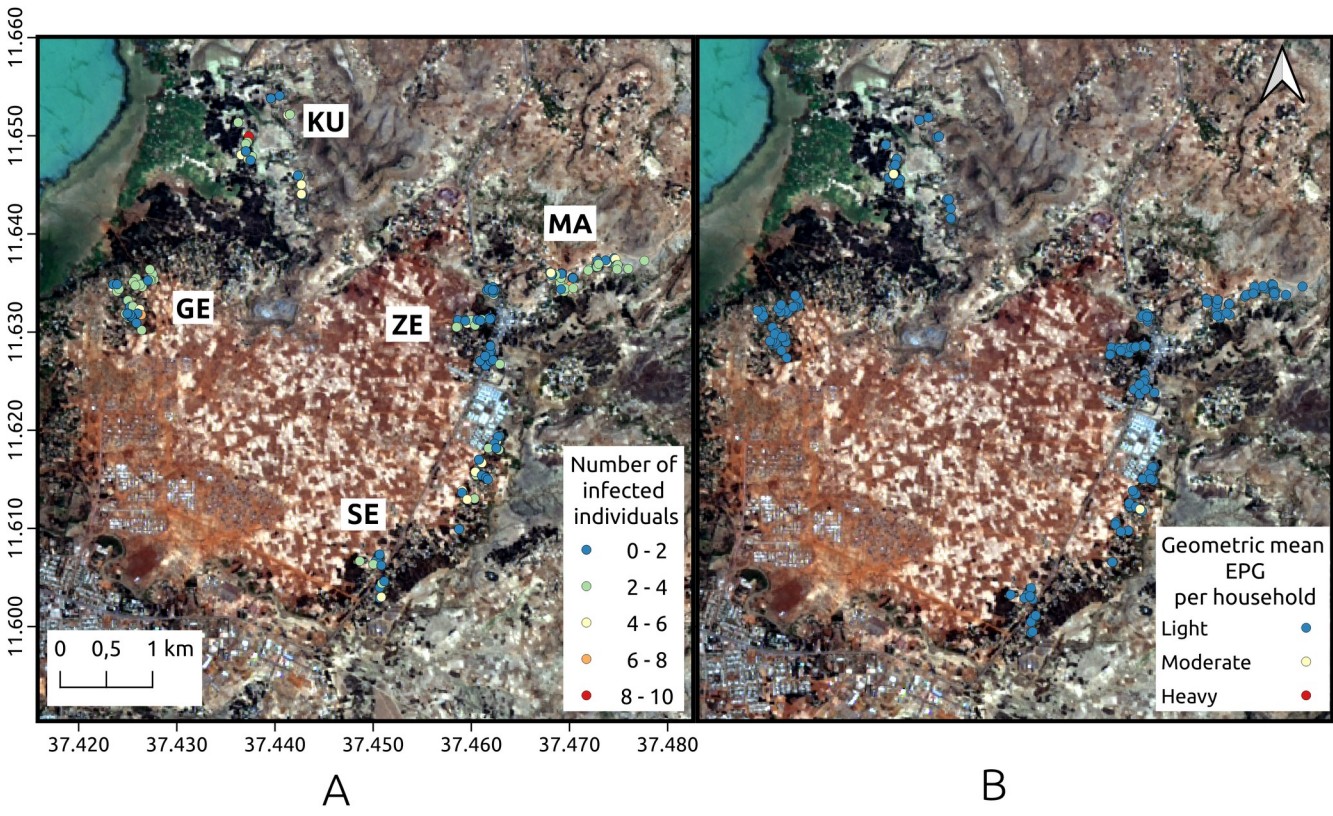

**Fig 3. Spatial distribution of hookworm infection.** A. Number of infected individuals per household. B. Geometric mean EPG in each of the households for the five villages. GE: Gedro, KU: Kurbi, MA: Mazoria, SE: Sesaberet, ZE: Zenzelema. Landsat-8 images, "Image courtesy of the U.S. Geological Survey", downloaded from the US Geological Survey server (USGS, https://www.usgs.gov) using Earth Explorer platform (https://earthexplorer.usgs.gov). The geographical location for Landsat-8 is: Path 170, Row 52. DATE_ACQUIRED = 2016-03-10.

households presented at least three infected individuals (Fig 3A) and the burden of infection was low, with more than 80% of the participants harbouring light hookworm infections and very few households with individuals harbouring moderate or high intensity infections (Fig 3B). Kurbi, Gedro and Sesaberet had one household each with moderate mean geometric EPG for the entire household. On the other hand, none of the villages contained households with high intensity geometric mean EPG. Moreover, in Zenzelema, none of the households had more than five infected individuals and all of the average hookworm infections were of light intensity. According to the spatial cluster analysis, using the normal Statscan model, one cluster with high values for mean geometric EPG was found in Sesaberet while high value clusters for the number of infected individuals were found in Mazoria, Gedro and Kurbi. The spatial localization of these clusters is observed in Fig 4.

The spatial distribution found was in agreement with what was observed in the terrain, with clear differences between the villages, as already described in the method section, but especially between Zenzelema and the rest of the villages. The habitat surrounding the houses was clearly different and more appropriate for the development of larvae in Mazoria than in Zenzelema as can be observed in Fig 3.

## Household Socioeconomic characterization

The database generated consisted of 138 households with complete socioeconomic characterization corresponding to the villages of Sesaberet, Mazoria and Zenzelema. The average

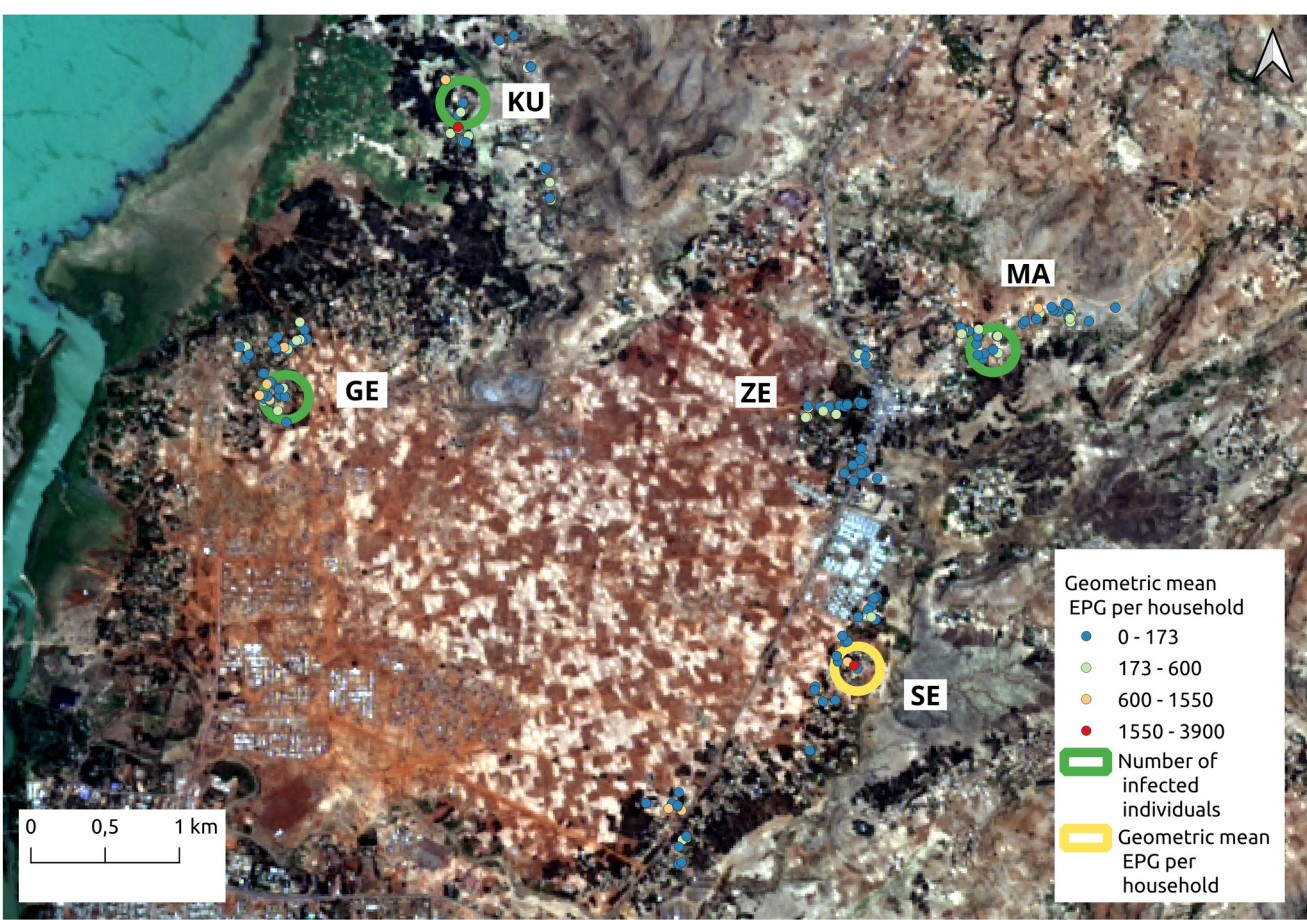

**Fig 4. Spatial clusters of households with high value of geometric mean EPG per household and number of infected individuals.** The color of each household point refers to the geometric mean EPG value. GE: Gedro, KU: Kurbi, MA: Mazoria, SE: Sesaberet, ZE: Zenzelema. Landsat-8 images, "Image courtesy of the U.S. Geological Survey", downloaded from the US Geological Survey server (USGS, https://www.usgs.gov) using Earth Explorer platform (https://earthexplorer.usgs.gov). The geographical location for Landsat-8 is: Path 170, Row 52. DATE_ACQUIRED = 2016-03-10.

number of inhabitants per household was four. These 138 households were all the same with respect to the roof, floor and wall characteristics. All the roofs were made of corrugated iron, all the floors were made of straw pasted with mud/cow dung and all the walls were made of wood and mud with straw. S1 Table describes the socioeconomic variables collected for each household. A total of 212 households participated in the simple household questionnaire and a sub-group of 138 household participated in the extended questionnaire which included individual information on the head of the household. The relation between the level of EPG with respect to all the measured socioeconomic variables was explored. Fig 5 shows the graphical representation using box plots only of those variables which showed statistical significance with respect to the level of EPG observed (p<0.05).

As observed in Fig 5, those individuals which practiced open defecation had a significant higher number of hookworm eggs in their stool. This was also true for those that did not have electricity, and those that used water from treated boreholes for cooking and handwashing. Moreover, adults showed statistically significant higher hookworm egg counts than children. Adults that own their own land also had significant higher eggs counts.

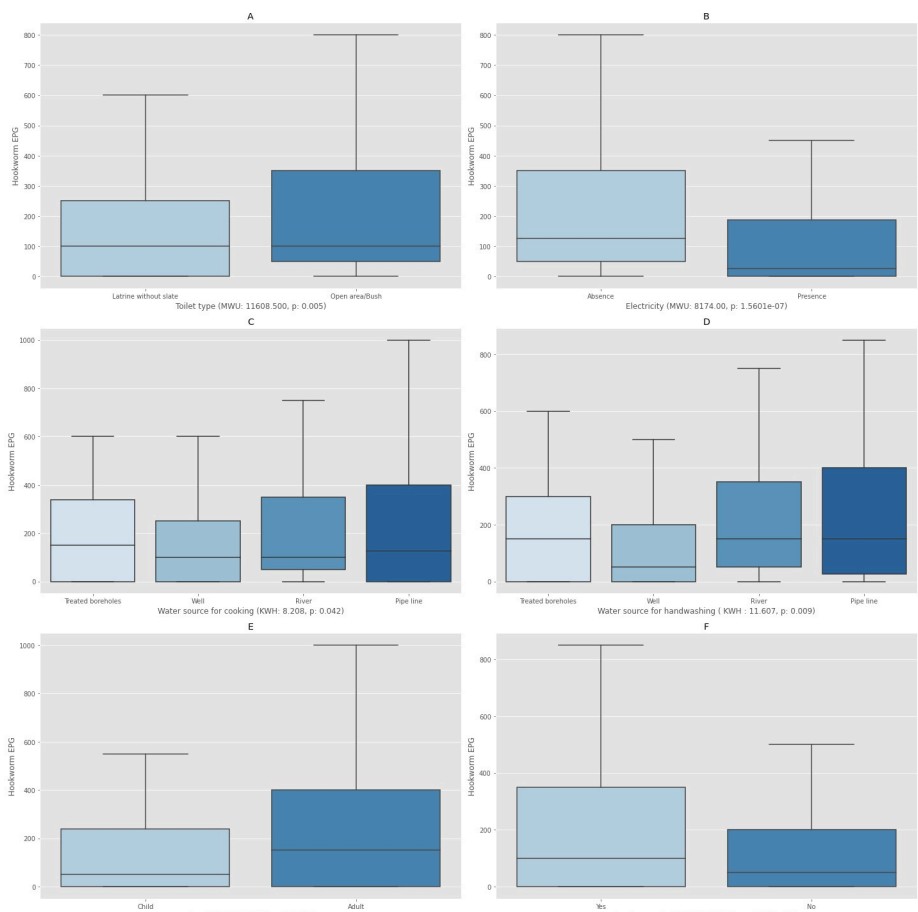

**Fig 5. Boxplots showing the association between the individual number of eggs per gram of feces (EPG) and characteristics of the households.** A. Toilet Type. B. Presence/absence of electricity. C. Water source for cooking. D. Water source for handwashing. E. Age (Child refers to all those under 14 and Adult refers to all those 15 or older). F. Land property. MWU: Mann-Whitney U, KWH: Kruskal-Wallis H, p: significance test (p<0.05).

## Spatial distribution of environmental and soil characteristics

The spatial distribution of some of the analysed soil and environmental indexes evaluated in the regions comprising all five villages is shown in Fig 6. The different characteristics were quite uniform between the villages, including altitude, although the area between Kurbi and Gedro has higher vegetation and therefore the values for the normalized difference vegetation index (NDVI) were high, while the values for the moisture stress index (MSI), bare soil index (BSI) and normalized difference built-up index (NDBI) were low. This makes sense since NDVI and MSI are associated with vegetation content and hydric stress while BSI and NDBI are associated with urban areas and bare soil or fallow lands. Taking a closer look by village, Gedro, Sesaberet and Kurbi had the highest average values for NDVI around the households (0.31, 0.27 and 0.27 respectively), while the lower average values were found in Zenzelema (0.23) and Mazoria (0.23), while both Gedro and Zenzelema had higher average values of MSI around the households (1 and 0.93, respectively). Mazoria also had a high average value of BSI around the households (-0.017), while both Gedro and Zenzelema had low values (-0.35 and -0.33, respectively). The average NDBI index was higher in Zenzelema (-0.008) and Mazoria (-0.004) with respect to the rest of the villages (Gedro -0.035, Kurbi -0.023 and Sesaberet

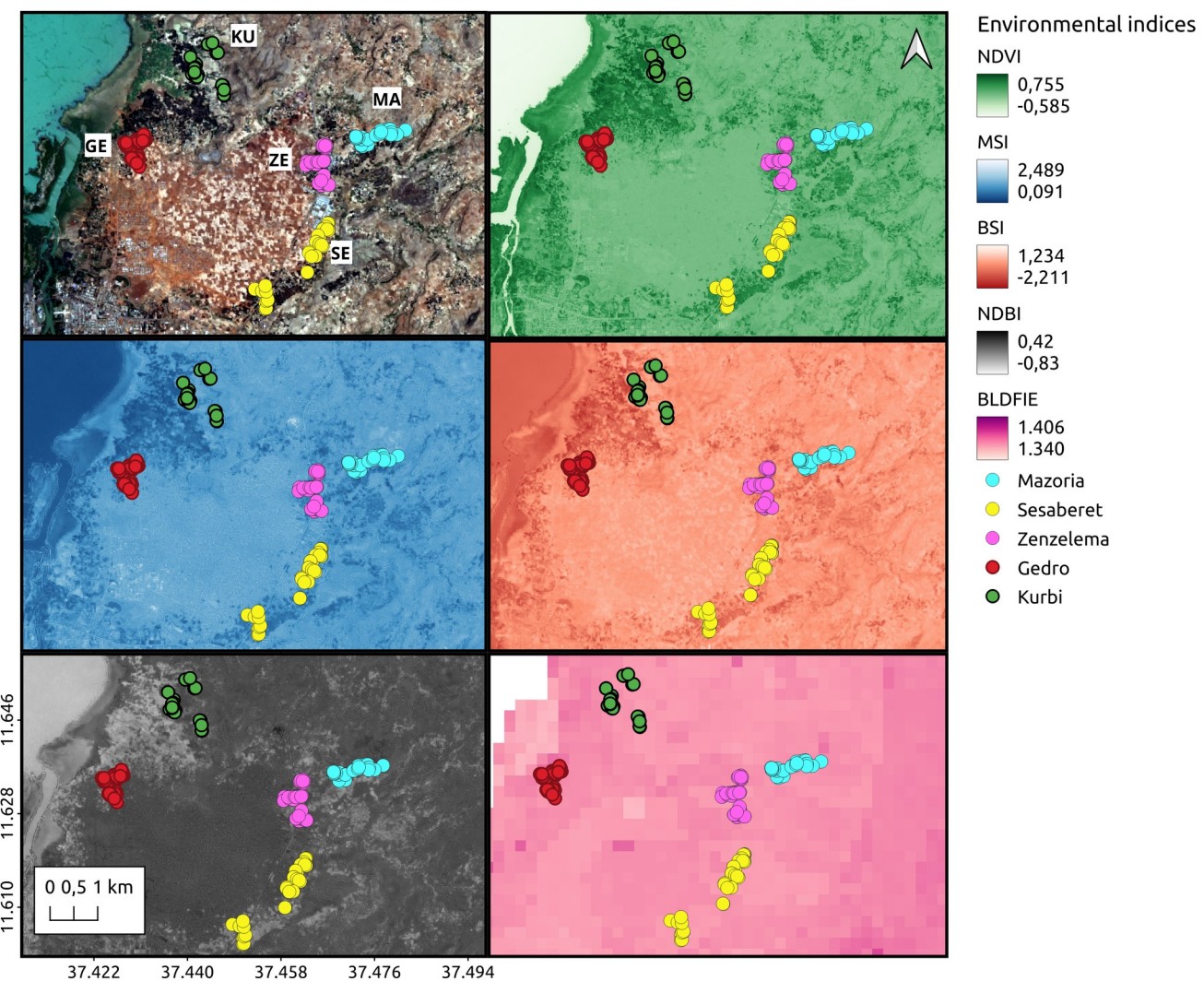

**Fig 6. Spatial distribution of the different soil and environmental indexes analyzed.** NDVI: Normalized Difference Vegetation Index. MSI: Moisture Stress Index. BSI: Bare Soil Index. NDBI: Normalized Difference Built-up Index. BLDFIE: Soil Bulk density. Landsat-8 images, "Image courtesy of the U.S. Geological Survey", downloaded from the US Geological Survey server (USGS, https://www.usgs.gov) using Earth Explorer platform (https://earthexplorer.usgs.gov). The geographical location for Landsat-8 is: Path 170, Row 52. DATE_ACQUIRED = 2016-03-10.

-0.022) and the average Dry bulk density index (BLDFIE) was highest in Sesaberet (1359.83) and lowest in Zenzelema (1320.18).

Taking the different parameters together, along with the infection characteristics, Kurbi had a high cluster of infected individuals in households that are located in an area with a high vegetation and hydric stress index. Mazoria also had a high cluster of infected individuals in households that were closer to an area with vegetation, higher hydric stress but also some bare soil. On the other hand, Gedro and Sesaberet had higher vegetation cover, higher bulk density and lower hydric stress and construction index, although Gedro had a high cluster of infected people while Sesaberet did not. Gedro also had a high cluster of households with high mean EPG and both of these clusters were located in households that were closer to an area with more bare soil. The households in Zenzelema presented surroundings with lower vegetation cover, lower proportion of bare soil and lower bulk density with higher hydric stress and construction index (BSI), and it also presented a high cluster of mean EPG per household.

Sesaberet was the only village with no clusters of infection or high mean EPG and the households were surrounded by more vegetation and higher bulk density.

## Household risk of hookworm infection

Only 89 households had complete socioeconomic data (all variables in the questionnaire were obtained). These households correspond to the villages of Sesaberet (SE), Mazoria (MA) and Zenzelema (ZE), thus including 368 individuals. This smaller database was used to model the risk of infection (number of individuals infected by hookworm per household–Ind_Numb). The model obtained through the linear multiple regression analysis showed that only the following variables significantly contributed to the model ($p<0.05$): BSI, MSI, Ind_Numb, and Soil organic carbon concentration (ORCDRC) ($R^2$: 0.756, AIC: 229.4, the coefficients were 3.4041, -3.7972, 7.7182, and 1.1548 respectively). Given only environmental characteristics and none of the socioeconomic characteristics from the extended questionnaire significantly contributed to the model, all the households with simplified questionnaires were included in order to broaden the database and explore the risk of infection in the entire study area. Accordingly, the model was run again with the 218 georeferenced households from the five villages which included 727 individuals. The linear multiple regression analysis ($R^2$: 0.767, AIC: 566.7) showed that only BSI (1.3208) and Ind_Numb (0.7361) remained significant, while MSI and ORCDRC were no longer significant. The following variables also significantly contributed to the model ($p<0.05$): temperature (0.5163), pH index measured in water solution (Phihox) (-0.8366), BLDFIE (0.0090) and NDVI (4.1564). BSI, Ind_Numb, temperature, BLDFIE and NDVI showed a positive association with the risk of infection, while Phihox showed a negative association.

In other words, if the land surrounding the household had denser and healthier surrounding vegetation, higher temperature, a greater surface of bare soil or fallow lands, and more compacted soils (higher bulk density), the probability of infection with hookworm increased. At the same time, the number of individuals present in a household was a determinant for the risk of hookworm infection, with the risk increasing as the number of individuals per household increased. On the contrary, the risk increased as the pH in the soil decreased, which means a more acidic than alkaloid pH would be favourable for the development of hookworm larvae in the soil.

Once the risk model for hookworm infected individuals per household was applied to the household taking into consideration the characteristics considered, the estimated number of infections per household was predicted and is presented in Fig 7A. There was a coincidence in the prediction with respect to the observed values (Fig 7B). The few households that presented a different estimation than the one observed, overestimated the number of infected individuals and in no cases did it underestimate the risk of infection.

## Discussion

The dominant STH found in all the villages was hookworm, except for Zenzelema where the dominant species was *S. stercoralis*, both of which are transmitted percutaneously. The other two STH species, *A. lumbricoides* and *T. trichiura*, which are transmitted by the faecal-oral route, were found at very low prevalences [8]. This is in agreement with what was found in other studies conducted in the area, with a predominance of hookworm in rural areas and a predominance of *A. lumbricoides* and *T. trichiura* in urban areas [6,7,24]. The village with the highest prevalence of hookworm, Kurbi, also had the highest average number of individuals per household infected and the neighbouring village of Gedro was the only village with high intensity hookworm infection. In this study, the concept that around 20% of individuals

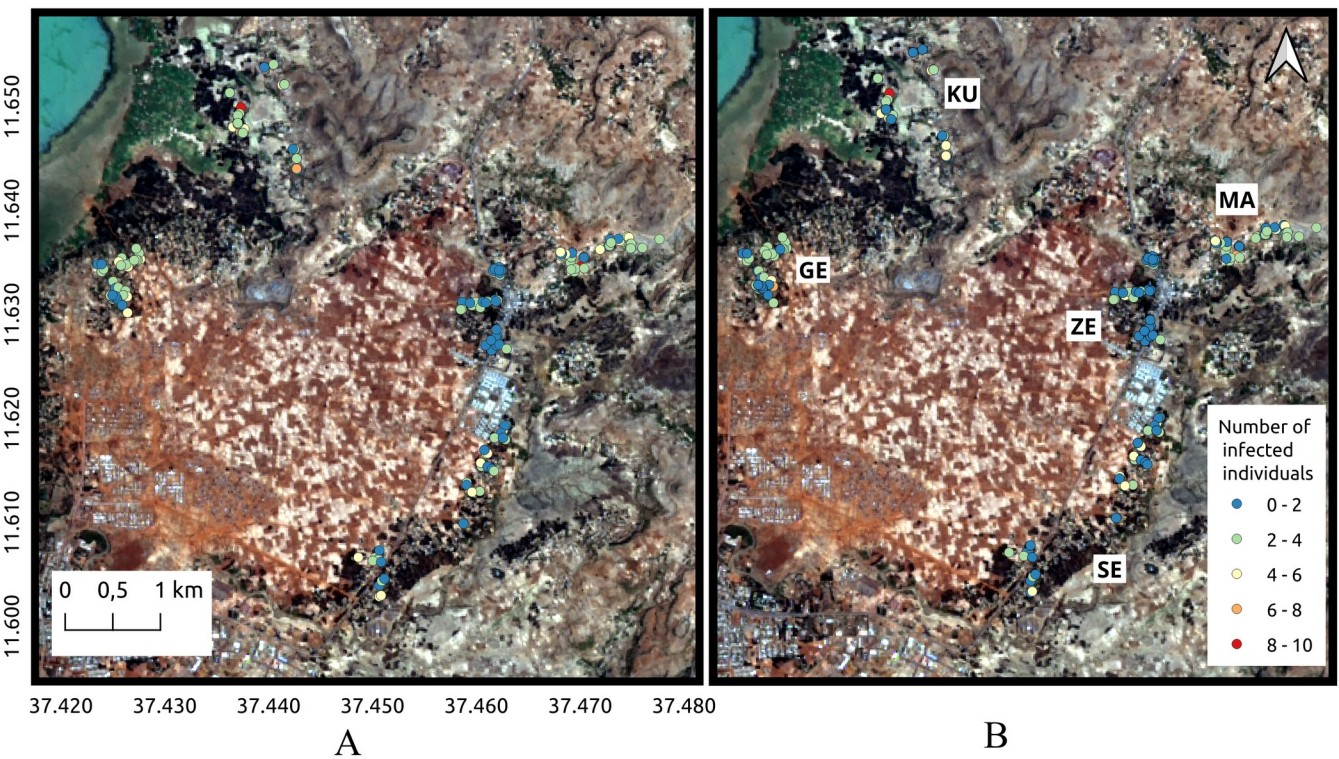

**Fig 7.** Map of the number of the hookworm infected individuals predicted (A) and observed (B) per household in the five villages. GE: Gedro, KU: Kurbi, MA: Mazoria, SE: Sesaberet, ZE: Zenzelema. Landsat-8 images, "Image courtesy of the U.S. Geological Survey", downloaded from the US Geological Survey server (USGS, https://www.usgs.gov) using Earth Explorer platform (https://earthexplorer.usgs.gov). The geographical location for Landsat-8 is: Path 170, Row 52. DATE_ACQUIRED = 2016-03-10.

harbour the higher intensity infections was confirmed [44], since around 80% of the infections found were of light intensity. These finding have implications on the control of STH in this area since adults act as reservoirs for infection for both hookworm and *S. stercoralis* [45]. Moreover, albendazole has been shown to be less effective as a single dose against hookworm [46] and is not effective against *S. stercoralis*, therefore the addition of ivermectin for MDA is recommended [47].

Both Kurbi and Gedro were the villages with the higher number of households with infected individuals with respect to the other three villages. Both of these villages were located closer to Tana Lake and have no access roads, therefore they are more remote than the rest of the villages included in the study. An area in Gedro with higher values for the number of infected individuals was identified; this might be due to the accumulation of infections over time and might be related to the way the health system is set up in the area. Normally, health extension workers (HEW) from the health centre in Zenzelema are in charge of preventive health activities in the villages, including house to house visits and controls in the health centre. The HEW are also in charge of MDA for PSAC but not for SAC since that program is implemented at the schools [10,11]. Therefore, access to the villages is important, with Zenzelema, Mazoria and Sesaberet having an easy access to the main road and to Bahir Dar City, while the situation of Kurbi and Gedro is challenging, especially in the rainy season, when lands are flooded and small rivers make access more difficult. In these scenarios, HEW are not able access these villages and the inhabitants themselves usually do not travel to the health centre in Zenzelema or to the hospital in Bahir Dar unless they are very sick. In this area, MDA

with albendazole is scheduled every 6 months, in May and November [12]. However, data on the coverage of the program is poor and punctual situations alter the schedule frequently (i.e. due to the SARS-CoV-2 pandemic, MDA was implemented in December 2020 in a house by house manner).

Gedro and Sesaberet were the villages with the greatest vegetation cover, soil bulk density and less hydric stress and construction index. On the contrary, Zenzelema had fewer infected individuals per household and lower intensity infections. The houses in this village presented lower vegetation cover, lower proportion of bare soil and lower soil bulk density, although they presented higher hydric stress and construction index. This is in agreement with what was observed in the field since Zenzelema is located near the main road that connects this area with the North of the country and it is the most urban of all the five villages included, with the houses adjacent to each other and with less vegetation cover. The area around the households in Mazoria was more densely vegetated, with covered soil that tends to retain certain moisture and shade. In an area like Mazoria or Gedro, more fields for cultivation were observed serving as cover for the practice of open defecation than in a setting like Zenzelema.

Individuals which practiced open defecation and those that owned their land for farming had a significant higher number of hookworm eggs in their stool; this relationship between hookworm infection and farming has been previously described [44,48–51] and is probably related to the age profile of hookworm infection; with an increase in prevalence and intensity over time which is not observed for *A. lumbricoides* and *T. trichiura* [45]. Additionally, in the current study, those individuals that did not have electricity as well as those that used water from treated boreholes for cooking and handwashing also had a significant number of hookworm eggs in their stool. Unsafe water has also been previously associated with STH infection [19,20,52]. As part of the prevention activities performed by HEW in the different villages included in this study, are those related to the promotion of treatment of boreholes with chlorine. The finding that treated boreholes are associated with greater hookworm infection should be noted in order to revise the treatment of boreholes with respect to correct dosing of chlorine, safe storage/handling of water at the household level and the existence of a fence or cover to protect the borehole and avoid recontamination. The placement of the latrine and borehole should also be taken into consideration to avoid contamination of the water with faecal material.

The relationship between STH and a lack of sanitation and hygiene has been previously reported in Ethiopia with both positive [20,24,48,52–54] and negative associations [55]. Moreover, the relationship between a lack of electricity and the presence of intestinal parasite infection has also been found in other studies [21,56]. The association between intensity of hookworm infection and the use of borehole water or lack of electricity is shown as significant in this study area and thus a lack of safe water for proper hygiene or lack of electricity to avoid stepping on unwanted remains could be playing a role in re-infection and therefore in the accumulation of infection, but the causality of the association cannot be established. Additionally, adults showed statistically higher hookworm egg counts than children; this confirms that for hookworm, adults tend to have higher intensities of infection contrary to what is observed for *A. lumbricoides* and *T. trichiura* [57]. Moreover, adults are not usually targeted for STH MDA so they may accumulate the infection over time and continue contaminating the environment. Although the intensity of infection was influenced by the sanitary conditions of the household, the probability of infection with hookworm did not seem to be determined by socioeconomic conditions [23], except for the number of individuals per household, since infection tends to be more prevalent when there's overcrowding [19,22,58,59].

Nonetheless, infection does seem to be related to the environmental conditions around the household. The infection was more probable in households surrounded by a combination of

vigorous vegetation and bare soil, high temperatures and compacted soils (high bulk density) with more acidic pH [16,18,27,60,61]. This coincides with the findings of this study, since Gedro was the village that presented the highest number of infected individuals and higher intensity infections and was located in an area with greater spatial vegetation cover, compacted soil and lower hydric stress and construction index. This has already been observed in other studies with respect to the precipitation, temperature and vegetation cover [14,62,63] but also with respect to the pH since hookworms can tolerate a pH range of 4.6 to 9.4 were they are still able to hatch and infect [64]. The presence of vigorous vegetation also favours the act of open defecation since it gives the individual privacy. Several studies have analysed the determinants associated with the practice of open defecation [65–67], some of which might also be playing a role in our study area, such as rurality, vegetation cover, nearness to water bodies, availability and type of latrine, and financial constraints.

With respect to the analysis of household risk for hookworm infection, a coincidence in the prediction is observed, with a lack of an underestimation of the risk, even though a few households manifest an overestimation of the number of individuals infected. Although the value of precision of the model is acceptable, it would be interesting to analyse how the incorporation of the predictive variables of different factors and different spatial resolutions modify the performance of the prediction. The model generated would allow predicting the numbers of infected individuals per household in the study area as more predictive variables are added and updated. The estimated variables obtained from satellites are easily updatable in a frequent manner, while the variables obtained in the field (i.e. number of infected individuals per household) are relatively static. Therefore, a probability infection map that is frequently updated may be obtained so as to guide the health authorities in order to be able to plan actions for the prevention and control of hookworm and other soil-transmitted helminths. Certain limitations of the study might have influenced the results, since not all nine villages from the area were included in the study and not all houses participated in the extended household questionnaire. Other limitations include the diagnostic techniques used since PCR was used only for detection of *S. stercoralis* [8} and not the other species of STH, knowing the species of hookworm involved might have provided information on the transmission route (if only percutaneous or if oral transmission might also be occurring, in the case of *Ancylostoma duodenale*).

## Conclusion

The use of environmental variables with high resolution, with a 30-meter radius around the household, together with the hookworm infection and intensity data obtained from the field [8], evidenced that for the establishment of hookworm infection, a suitable environment is necessary. Based on previous studies and the one presented herein, a suitable environment is one where there is precipitation to provide humidity as well as vegetation to provide shade, and a soil with appropriate organic content and density to avoid larvae desiccation. Moreover, once the infection has been established, sociodemographic, socioeconomic and behavioural parameters (such as age, source of water or practice of open defecation) seem to then play a role in the maintenance of infection as well as its intensity. These findings will aid in the identification of hookworm risk areas and also to guide program managers to consider expanding MDA programs to include adults and integrate activities with the water, sanitation and hygiene (WASH) sector.

## Supporting information

**S1 Table. Socioeconomic characteristics obtained using the extended questionnaire by village (Mazoria, Zenzelema and Sesaberet).**
(DOCX)

## Acknowledgments

We would like to thank the Amhara National Regional State Health Bureau in Bahar Dar for its collaboration and support in this study. We appreciate the support of the Zenzelema Health Center director Mr. Tadesse Meseret Kokeb and all the staff for their collaboration. We are most grateful to the community leaders for facilitating the participation and contact with the community. We would also like to thank Marcelo Abril and Marcelo Scavuzzo, directors of Fundación Mundo Sano and Instituto Gulich, respectively, for their institutional support.

## Author Contributions

**Conceptualization:** Aranzazu Amor Aramendia, Zaida Herrador, Maria Victoria Periago.

**Data curation:** Melaku Anegagrie, Sofía Lanfri, Aranzazu Amor Aramendia, Carlos Matías Scavuzzo.

**Formal analysis:** Aranzazu Amor Aramendia, Carlos Matías Scavuzzo, Maria Victoria Periago.

**Investigation:** Aranzazu Amor Aramendia, Zaida Herrador, Agustín Benito, Maria Victoria Periago.

**Methodology:** Sofía Lanfri, Aranzazu Amor Aramendia, Carlos Matías Scavuzzo, Zaida Herrador, Agustín Benito, Maria Victoria Periago.

**Project administration:** Melaku Anegagrie.

**Software:** Sofía Lanfri.

**Supervision:** Aranzazu Amor Aramendia, Maria Victoria Periago.

**Writing – original draft:** Melaku Anegagrie, Sofía Lanfri, Maria Victoria Periago.

**Writing – review & editing:** Aranzazu Amor Aramendia, Zaida Herrador, Agustín Benito, Maria Victoria Periago.

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
