## [Decision Letter · Decision Letter 0]

25 Mar 2021

Dear Dr. Periago,

Thank you very much for submitting your manuscript "Environmental characteristics around the household and their association with hookworm infection in rural communities from Bahir Dar, Amhara Region, Ethiopia" for consideration at PLOS Neglected Tropical Diseases. As with all papers reviewed by the journal, your manuscript was reviewed by members of the editorial board and by several independent reviewers. In light of the reviews (below this email), we would like to invite the resubmission of a significantly-revised version that takes into account the reviewers' comments. 

We cannot make any decision about publication until we have seen the revised manuscript and your response to the reviewers' comments. Your revised manuscript is also likely to be sent to reviewers for further evaluation.

Sincerely,

Subash Babu

Associate Editor

Robert Reiner

Deputy Editor

Reviewer's Responses to Questions

**Key Review Criteria Required for Acceptance?**

**Methods**

-Are the objectives of the study clearly articulated with a clear testable hypothesis stated?

-Is the study design appropriate to address the stated objectives?

-Is the population clearly described and appropriate for the hypothesis being tested?

-Is the sample size sufficient to ensure adequate power to address the hypothesis being tested?

-Were correct statistical analysis used to support conclusions?

-Are there concerns about ethical or regulatory requirements being met?

Reviewer #1: Methods 

Please add subheading to define or describe the terms like woredas???, and kebeles???

Data collection 

Line 127: “using WHO standardized surveys” I think this needs reference 

Line 129: ….. while the head of the family was also asked to answer a household questionnaire. I think it will be good to explain slightly this household question for the heads

Line 133: Stool samples were processed as previously described (6) to determine presence and intensity of hookworm infection. Actually, in ref 6 they used 3 three methods. IF you used also the 3 good to reported before referring to 6.

Reviewer #2: It is necessary to inform how the parasitological examinations were performed. Was only the McMaster technique performed? It is necessary to describe in detail the laboratory procedures used for the diagnosis of hookworm infections. 

One of the dependent variables used in the analyzes was the "mean number of EPG per household". The use of means requires information on standard deviation. It also requires that the distribution be normal. Justify the use and adequacy of the "mean number of EPG per household" as a parasitological parameter.

Reviewer #3: The method section provides no details of the laboratory analysis. How were specimens collected? How many specimens were collected? How long and in what way were they stored and processed prior to analysis? Where was analysis performed? What methodology was used (MacMaster flotation is mentioned in the footnotes of table 1, but no details of how this was performed have been provided)? What was the level of competence of the microscopists and was any quality control performed such as re-analysis of a percentage of samples by a separate technician? This is a major flaw and must be addressed thoroughly.

**Results**

-Does the analysis presented match the analysis plan?

-Are the results clearly and completely presented?

-Are the figures (Tables, Images) of sufficient quality for clarity?

Reviewer #1: they did good and required analysis 

Results 

Line 206 -208: “Stool samples were processed as previously described (6) to determine presence and intensity of hookworm infection. “ I will suggest to remove this from the results section to avoid any confusion.

Line 213-214: Table 4. Infection status of the participants of the study from the five villages, Kurbi, Gedro, Mazoria, Sesaberet and Zenzelema (Amhara State, Ethiopia). I will suggest the following “Hookworm infection status in study participants from the five villages, Kurbi, Gedro, Mazoria, Sesaberet and Zenzelema, Amhara State, Ethiopia

Table 4 needs also a formatting e.g presenting this in landscape 

Line 231-233: I will suggest removing this from the results section, will fit better to discussion 

Line 272: needs space after (BLDFIE)

Reviewer #2: In Table 4, in the line referring to No. of infected individuals by intensity as per MM [No.] , the sum of numbers in the intensity categories are sometimes not is equal to the total number of positives. In Table 4, in addition to presenting "Mean number

of infected individuals per household (mean)", it would be important to present the proportion of positive subjects among the total tested, per household.

Reviewer #3: The numbers in table 1 do not add up correctly and should be checked. Also, please use geometric mean instead of mean for average egg intensity counts.

**Conclusions**

-Are the conclusions supported by the data presented?

-Are the limitations of analysis clearly described?

-Do the authors discuss how these data can be helpful to advance our understanding of the topic under study?

-Is public health relevance addressed?

Reviewer #1: the manuscript needs a conclusion section which is missing in this current version

Reviewer #2: The correlation between farming and higher hookworm infection intensity should be better discussed. The permanence of hookworm infections despite the significant reduction in the prevalence of infections by Ascaris and Trichuris has been observed in several regions, in other countries, with the same characteristics. This finding, that is, the different epdemiological behavior of percutaneously and orally transmitted STHs needs to be better addressed in the discussion.

It is necessary to discuss the determinants of the practice of open defecation. Since this practice is closely related to the frequency and intensity of hookworm infections, it is necessary to understand it. Are there cultural determinants? Do people who practice subsistence farming need to defecate in the workplace? What alternatives to open defecation can be proposed to populations living in this socio-demographic and environmental scenario?

Reviewer #3: There is no final conclusion paragraph and this should be added.

The statistically significant associations of environmental factors with STH infection (e.g. borehole treatment) have been accepted as causative, but how they might be causative is not well addressed, could some of these associations be related to other factors which are actually causing the high rates of hookworm infection? 

A major area requiring attention is the depth of discussion. At present, it is very descriptive of the situation in the area of Ethiopia analysed, but does not infer how this might relate to other regions of the world affected by STH. 

What has been learned about risk factors and environment for STH? 

Can anything be learned from that about preferred environments for hookworm and Strongyloides (since these were the most common STH recovered)?

How does the data derived from this work compare to other STH studies performed elsewhere in the world or Ethiopia in particular? Were the findings presented a high or low prevalence and intensity of STH infections compared to other regions of Ethiopia?

The discussion also fails to address any laboratory methodological limitations of the study and avenues for further research. 

Would PCR have provided better results? 

How does MacMaster compare to the WHO standard method of Kat Katz? 

What about hookworm species – as Necator americanus has different levels of environmental resistance to Ancylostoma duodenale, and Ancylostoma ceylanicum has a zoonotic reservoir, would knowing the species of hookworm infecting people by using PCR have provided further information for analysis?

**Editorial and Data Presentation Modifications?**

Reviewer #1: Line 39 : more acidic pH this is vague please I will suggest to give a range or more precision 

Author summary 

Line 47 : I will suggest to precisely define the environment element that the authors assessed in their work 

Line 52-53: please revise to keep the

“intensity of hookworm infection was associated to certain socioeconomic conditions such as the practice of open defecation and a lack of electricity”

Key words: 

Line 57: I will suggest removing Soil-transmitted helminths

Introduction 

Line 61-62 and line 65: please add the abbreviation at first appearance Soil-Transmitted

Helminthiases 

Discussion 

Line 342: lack of electricity associated with STH infection need cautious interpretation because this can stand as confounders 

Also during the discussion I will suggest the authors to discussion the implication of their findings in the currents strategies against STH or Hookworm

Reviewer #2: (No Response)

Reviewer #3: Please see my comments in the attached .pdf

**Summary and General Comments**

Reviewer #1: Despite the ongoing mass drug administration there is bottle neck about the elimination STH in several countries and this type of assessment done by María Victoria Periago et al. is useful to explore the environmental characteristics and their implications on NTD.

This is very interesting areas of research on Neglected tropical diseases to study the impact of environment on the occurrence or presence of STH under ongoing mass drug administration with the goal to eliminate them in the near future. They came with very important findings showing that open defecation and lack of electricity, being adults are associated with hookworm infection. These findings add new knowledge in the field of NTDs research and will lead to reassess new strategies or improve existing strategies.

However, the authors should report also the few cases of STH before focusing on the hookworm found during their study.

Reviewer #2: This is an interesting study on the spatial distribution of STH in endemic areas in Ethiopia. The study has scientific merit and adds relevant information for the improvement of hookworm control strategies. The article describes the current scenario of eco-epidemiology of STHs, which is the same in different developing countries, in rural or peri-urban populations. This scenario is characterized by: i) permanence of hookworm infections, ii) frequent failure of mass drug administration policies due to re-infections and maintenance of infections in adolescents and adults, iii) permanence of populations practicing open defecation in peridomestic areas, iv) predominance of low parasitic loads. The recognition of this landscape is necessary for the improvement of control policies.Some methodological improvements were pointed out, as well as some aspects related to the interpretation of the data.

Reviewer #3: Please see my comments in the attached .pdf

PLOS authors have the option to publish the peer review history of their article (what does this mean?). If published, this will include your full peer review and any attached files.

Reviewer #1: Yes: Housseini Dolo

Reviewer #2: Yes: Filipe Anibal Carvalho Costa

Reviewer #3: Yes: Richard Bradbury
---

## [Decision Letter · Decision Letter 1]

11 May 2021

Dear Dr. Periago,

We are pleased to inform you that your manuscript 'Environmental characteristics around the household and their association with hookworm infection in rural communities from Bahir Dar, Amhara Region, Ethiopia' has been provisionally accepted for publication in PLOS Neglected Tropical Diseases.

Best regards,

Subash Babu

Associate Editor

Robert Reiner

Deputy Editor

Reviewer's Responses to Questions

**Key Review Criteria Required for Acceptance?**

**Methods**

-Are the objectives of the study clearly articulated with a clear testable hypothesis stated?

-Is the study design appropriate to address the stated objectives?

-Is the population clearly described and appropriate for the hypothesis being tested?

-Is the sample size sufficient to ensure adequate power to address the hypothesis being tested?

-Were correct statistical analysis used to support conclusions?

-Are there concerns about ethical or regulatory requirements being met?

Reviewer #2: The objectives test a clear hypothesis and the methodology is adequate. The revised version of the manuscript made the methodology clearer.

Reviewer #3: (No Response)

**Results**

-Does the analysis presented match the analysis plan?

-Are the results clearly and completely presented?

-Are the figures (Tables, Images) of sufficient quality for clarity?

Reviewer #2: The presentation of the results has been improved, making it clearer. The requested corrections were incorporated into this new version of the manuscript.

Reviewer #3: (No Response)

**Conclusions**

-Are the conclusions supported by the data presented?

-Are the limitations of analysis clearly described?

-Do the authors discuss how these data can be helpful to advance our understanding of the topic under study?

-Is public health relevance addressed?

Reviewer #2: The data presented support the conclusions and the importance in public health is clear.

Reviewer #3: (No Response)

**Editorial and Data Presentation Modifications?**

Reviewer #2: --

Reviewer #3: (No Response)

**Summary and General Comments**

Reviewer #2: --

Reviewer #3: Line 133: Grammar. Change "were stool samples were processed" to "in which stool samples were processed"

Line 419: Attribution of an emotion to an inanimate concept: please change "These findings hope to aid" to "These findings will aid"

PLOS authors have the option to publish the peer review history of their article (what does this mean?). If published, this will include your full peer review and any attached files.

Reviewer #2: **Yes: **Filipe Anibal Carvalho Costa

Reviewer #3: No

---

## [Editor Report · Acceptance letter]

8 Jun 2021

Dear Dr. Periago,

We are delighted to inform you that your manuscript, "Environmental characteristics around the household and their association with hookworm infection in rural communities from Bahir Dar, Amhara Region, Ethiopia," has been formally accepted for publication in PLOS Neglected Tropical Diseases.

Best regards,

Shaden Kamhawi

co-Editor-in-Chief

Paul Brindley

co-Editor-in-Chief
